# Protein S-palmitoylation regulates different stages of meiosis in *Schizosaccharomyces pombe*

Thanh-Vy Pham[1,2] , Wan-Yi Hsiao[1], Yi-Ting Wang[1], Shu-Dan Yeh[2] , Shao-Win Wang[1]

Posttranslational protein S-palmitoylation regulates the localization and function of its target proteins involved in diverse cellular processes including meiosis. In this study, we demonstrate that S-palmitoylation mediated by Erf2-Erf4 and Akr1 palmitoylacyltransferases is required at multiple meiotic stages in the fission yeast *Schizosaccharomyces pombe*. We find that S-palmitoylation by Erf2-Erf4 is required for Ras1 localization at the cell periphery to enrich at the cell conjugation site for mating pheromone response. In the absence of Erf2 or Erf4, mutant cells are sterile. A role of Akr1 S-palmitoylating the nuclear fusion protein Tht1 to function in karyogamy is identified. We demonstrate that S-palmitoylation stabilizes and localizes Tht1 to ER, interacting with Sey1 ER fusion GTPase for proper meiotic nuclear fusion. In *akr1*, *tht1*, or *sey1* mutant, meiotic cells, haploid nuclei are unfused with subsequent chromosome segregation defects. Erf2-Erf4 has an additional substrate of the spore coat protein Isp3. In the absence of Erf2, Isp3 is mislocalized from the spore coat. Together, these results highlight the versatility of the cellular processes in which protein S-palmitoylation participates.

## Introduction

Protein lipidation is a unique co- or posttranslational modification in which lipid moieties are covalently attached to proteins. There are at least six types of lipids including fatty acids, isoprenoids, sterols, phospholipids, glycosylphosphatidylinositol anchors, and lipid-derived electrophiles, which can be covalently attached to proteins (Chen et al, 2018). Lipidation markedly increases the hydrophobicity of proteins, leading to changes in their conformation, stability, membrane association, localization, trafficking, and binding affinity to their co-factors. Protein lipidation is involved in pathways that play critical roles in cell signaling to regulate protein functions linked to diseases such as neurological disorders, metabolic diseases, and cancers (Yeste-Velasco et al, 2015; Zaręba-Kozioł et al, 2018). It is important to understand the functions and regulatory mechanisms of protein

lipidation, which will advance our understanding of their pathological relevance, leading to strategies for targeting protein lipidation for therapeutic applications.

Fatty acylation is a type of protein lipidation with the attachment of saturated and unsaturated fatty acids to the glycine, serine, lysine, or cysteine residues of proteins. Palmitoylation and myristoylation represent the two most common forms of fatty acylation. Palmitoylation is a process involving the covalent attachment of the 16-carbon palmitate to proteins (Linder & Deschenes, 2007). Palmitoylation targets several protein residues, including serine (O-palmitoylation) and cysteine (S-palmitoylation or N-palmitoylation when it occurs at the N-terminal of the protein). Both O- and N-palmitoylation appear to be irreversible. Conversely, S-palmitoylation is reversible, and S-palmitoylated proteins can undergo cycles of acylation and deacylation in response to upstream signals (Mumby, 1997; Chen et al, 2018). In mammalian cells, S-palmitoylate is added by a family of 23 transmembrane zinc finger DHHC (Asp-His-His-Cys)–containing protein acyl transferases and is removed by fatty acyl protein thioesterases (Malgapo & Linder, 2021). About 10% of eukaryotic proteins are supposed to be S-palmitoylated, identified by high-throughput screens (SwissPalm database) (Blanc et al, 2015).

The reversible nature of S-palmitoylation enables fine-tuned regulation of protein function. More importantly, the proper function of many membrane proteins such as surface receptors and signaling proteins requires palmitoylation-mediated partitioning into lipid rafts. The small GTPase Ras, which is involved in cellular signal transduction, has been described in many studies as a palmitoylated protein, and site-specific palmitoylation affects its trafficking and domain localization (Onken et al, 2006). Given the vital role of signal transduction in cancer development (mutations in *RAS* genes are associated with 30% of human cancer) (Gimple & Wang, 2019), a connection of protein S-palmitoylation to cancer development is suggested. In keeping with this notion, the expressions of some zDHHC genes are altered in various cancer tissues (Ko & Dixon, 2018), but how these palmitoylacyltransferases participate in cancer development remains to be defined.

---

[1]Institute of Molecular and Genomic Medicine, National Health Research Institutes, Zhunan Town, Taiwan   [2]Department of Life Sciences, National Central University, Taoyuan, Taiwan

Correspondence: shaowinwang@nhri.edu.tw

The association of S-palmitoylation with cellular differentiation in different model organisms has also been reported. In *Arabidopsis thaliana*, male and female gametogenesis both require a fully functional AtPAT21 palmitoylacyltransferase (Li et al, 2019). *Xenopus* palmitoylacyltransferase ZDHHC3 plays an important role in oocyte maturation (Fang et al, 2016). The fission yeast *Schizosaccharomyces pombe* is a well-established model to probe the processes of sexual differentiation. Although a function of protein S-palmitoylation in the fission yeast meiosis has been suggested (Onken et al, 2006; Zhang et al, 2013), a detailed characterization of the involvements of these palmitoylacyltransferases in meiosis has not been described. In this study, we demonstrated that protein S-palmitoylation was required at different stages of meiosis in *S. pombe* and the underlying molecular mechanisms were determined.

## Results

### The fission yeast palmitoylacyltransferases

There are five palmitoylacyltransferases Akr1, Erf2, Pfa3, Pfa5, and Swf1 encoded in the *S. pombe* genome (Zhang et al, 2013). Each contains the conserved Asp-His-His-Cys (DHHC) motif in the enzyme active site with two to three transmembrane domains (Politis et al, 2005) (Fig 1A). Erf4 is an accessory protein of Erf2 (Salaun et al, 2020). Akr1 contains six additional ankyrin repeats for protein–protein interaction. As a first step toward the characterization of these enzymes, experiments were performed to determine their cellular localization. Given the relatively low expression level of these proteins (Harris et al, 2022), we generated C-terminally tagged GFP fusion protein, ectopically expressed by the thiamine-repressible *nmt1* promoter. To avoid ectopic localization of over-expressed proteins, we took advantage of the leaky *nmt1* promoter to moderately express these proteins in the presence of a low concentration of thiamine (Fig S1). In keeping with the idea of multidomain membrane-integrated proteins, examination of living cells by fluorescence microscopy revealed distinct localization patterns of these proteins throughout the secretory pathway. Interestingly, some showed specific subcellular localization such as Pfa3 and Pfa5 in the vacuole-like structures and Swf1 on the endoplasmic reticulum (ER: peripheral and nuclear membranes) (Fig 1B), whereas others appeared in more than one location. Akr1 proteins were found on the ER (Fig 1C) and mobile dots resembling Golgi (Fig 1D), and Erf2-Erf4 appeared in the cell periphery and Golgi. The localization of these proteins was essentially identical to those when cultured in the absence of thiamine expressed at the full strength of the *nmt1* promoter (Fig S2). The different pattern of cellular localization implies that these enzymes regulate the S-palmitoylation of substrates involved in diverse cellular processes. Except for Erf2-Erf4, the functionalities of these proteins in *S. pombe* have not been described. Experiments are under way to identify their substrates to determine the functionality of these proteins.

To gain more insight into the biological functions of these enzymes in *S. pombe*, the one-step gene disruption method was used

to delete the whole ORF of these palmitoylacyltransferases except for the essential *swf1* gene. In agreement with the PomBase genomic data (Harris et al, 2022), all mutants have been successfully obtained as validated by PCR and were viable. During the construction of these mutants, we observed that some mutants caused severe meiotic defects and were subjected to further phenotypical analysis using Hht2 (histone H3) GFP fusion protein as the nuclear marker. As shown in Fig 1E, we found that although other mutants proceeded through meiosis to generate spores after mating, mutants of Erf2 and its cofactor Erf4 failed to generate any zygote for spore formation and were completely sterile homothallic (self-conjugation) or heterothallic with WT cells. In addition, aberrant meiosis with unequal and inaccurate spore formation was also observed in the *akr1* mutant. Together, these results suggested that protein S-palmitoylation has multiple functions in meiosis. Experiments were performed to determine the underlining molecular mechanisms.

### Palmitoylation of Ras1 by Erf2-Erf4 is required for mating pheromone response

The results described above suggested a function of Erf2-Erf4 in zygote formation. Ras1 and Rho3, two substrates of Erf2-Erf4, have been implicated to function in meiosis (Onken et al, 2006; Zhang et al, 2013). Experiments were performed to determine their roles in meiosis. As shown in Fig 2A, similar to *erf2* and *erf4* mutants, the *ras1* mutant was defective in forming zygote. In contrast, although with reduced efficiency, *rho3* mutant can still progress through meiosis, forming asci. These results suggested that Ras1, but not Rho3, was the primary target of Erf2-Erf4 in meiosis and was explored further in the following experiments.

The Ras1-MAPK signaling pathway is intimately involved in controlling mating in *S. pombe* (Chang et al, 1994). Activation of Ras1 is essential to transduce the pheromone signal. It has been recently demonstrated that during early mating, Ras1-GTP exhibits oscillatory polarization dynamics (Merlini et al, 2018). Experiments were performed to determine whether palmitoylation of Ras1 by Erf2-Erf4 was required for these processes. As shown in Fig 2B, in mating *S. pombe* cells, GFP-Ras1 was enriched at the polarized projection, marked by myo52-tdTomato, over a broader zone. In the absence of Erf2, Ras1 was mislocalized from the cell periphery, forming granule-like structures in the cytoplasm. In contrast, no significant change in cellular localization of GFP-Rho3 was found during early mating or in the absence of Erf2. Erf2 S-palmitoylated Ras1 at cysteine residue 215 (Onken et al, 2006). As in the *erf2* mutant, Ras1 with cysteine residue 215 replaced by alanine residue was mislocalized from the cell periphery, forming granule-like structures in the cytoplasm (Fig 2C), and, unlike the WT Ras1, failed to rescue the meiotic defect of the *ras1* mutant that was revealed using Htb1 (histone H2B) mono cherry fusion protein as the nuclear marker (Fig 2D). Together, these results suggested that palmitoylation of Ras1 at Cys215 by Erf2-Erf4 was required for mating pheromone response to generate the zygote for spore formation. In addition to mating upon meiotic entry, Ras1 is also required for cell shape control during the vegetative growth phase. However, we found that in contrast to the pear-shaped vegetative *ras1* cells, *erf2* and *erf4* mutants displayed relatively normal cell morphology (Fig 2E). These results suggested

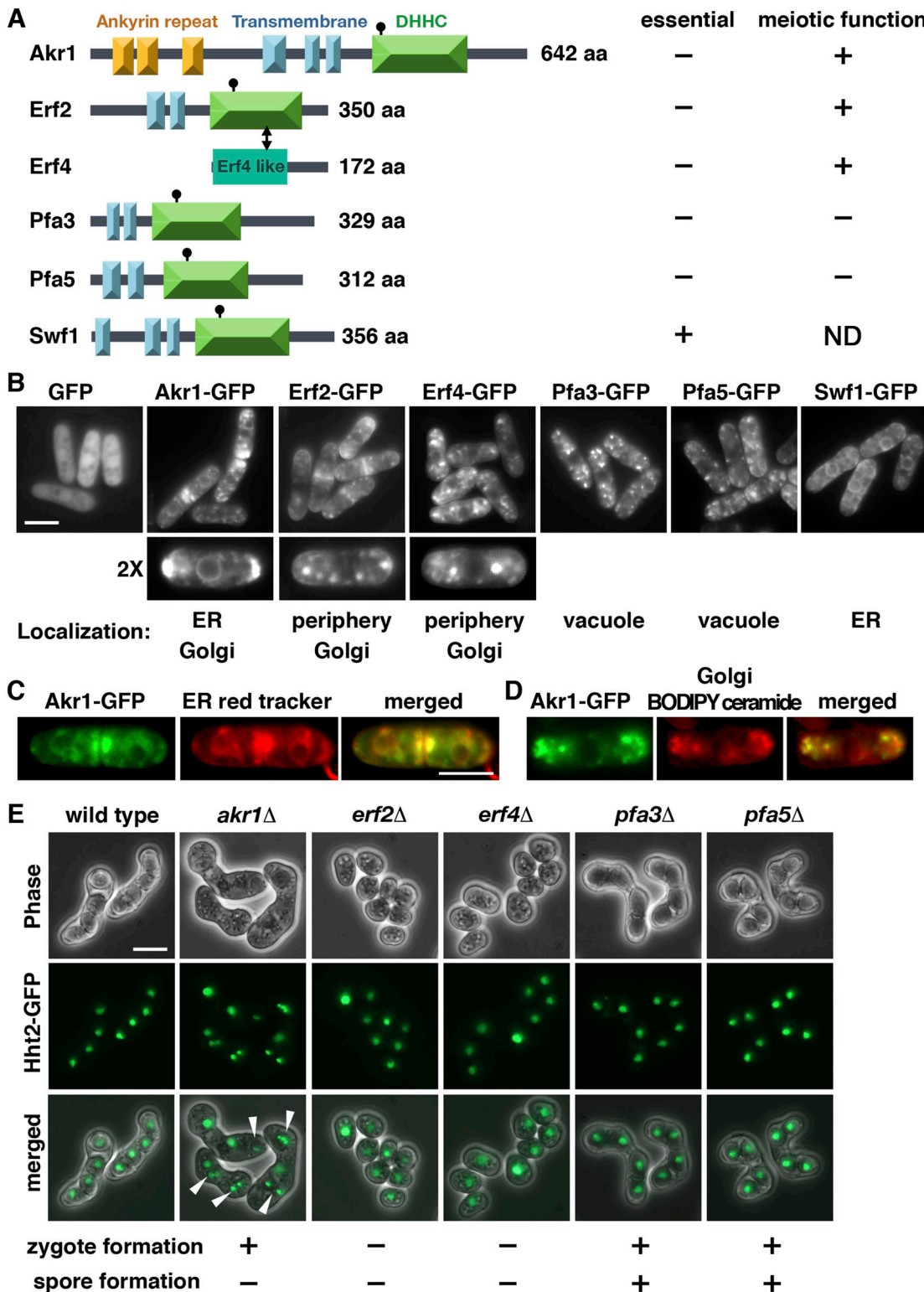

Figure 1. Fission yeast palmitoylacyltransferases.
**(A)** Schematic representation of the domain structures of the palmitoylacyltransferases in *S. pombe*. **(B, C, D)** Cellular localization of the fission yeast palmitoylacyltransferases. GFP fusion proteins with (C) ER red tracker and (D) BODIPY ceramide for Golgi structure applied to the study. Scale bar, 5 μm. **(E)** Micrographs of meiotic cells of the indicated homothallic strains expressing Hht2-GFP protein as the nuclear marker. Arrowheads indicate aberrant spores.

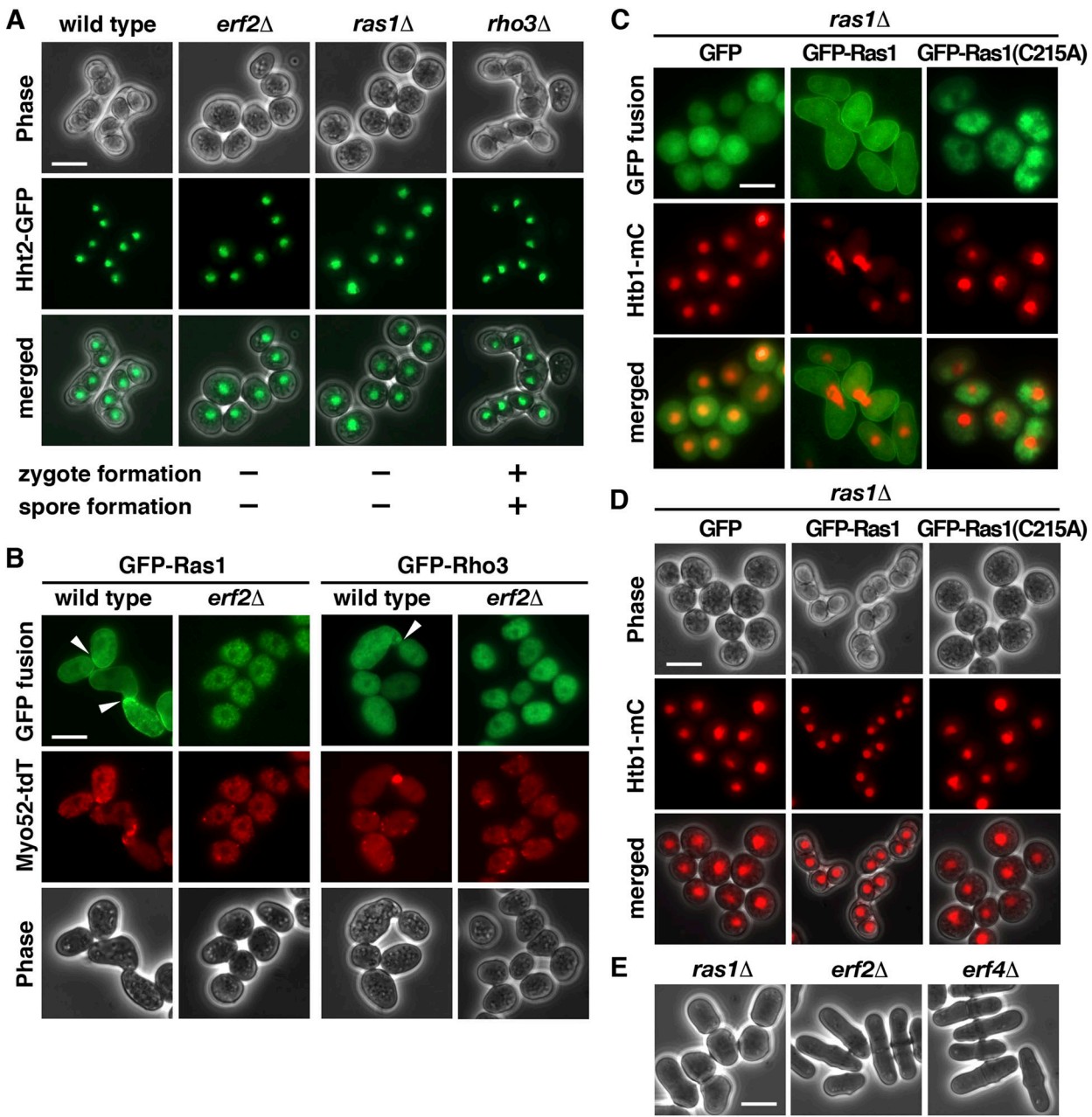

**Figure 2. S-Palmitoylation of Ras1 by Erf2-Erf4 is required for mating pheromone response.**
**(A)** Micrographs of meiotic cells of the indicated homothallic strains expressing Hht2-GFP protein as the nuclear marker. Scale bar, 5 $\mu$m. **(B)** Fluorescence micrographs of the indicated homothallic strains expressing GFP-Ras1 or Rho3 (green) also expressing Myo52-TdTomato (red) proteins. Arrowheads indicate polarized projections. **(C)** Fluorescence micrographs of *ras1* strains transformed with a plasmid expressing WT GFP-Ras1 or C215A mutant (green) and also expression of Htb1-mCherry (red) proteins. **(D)** Micrographs of homothallic *ras1* cells transformed with a plasmid expressing WT Ras1 or C215A mutant. Htb1-mCherry proteins were used as the nuclear marker. **(E)** Micrographs of vegetative cells of the indicated strains were shown.

that the requirement of Er2-Erf4 for Ras1 function seems only during meiosis.

### *akr1* mutant is defective in meiotic nuclear fusion

The mutant phenotype of *akr1* suggested a function in meiosis. To gain more insight into the role of Akr1 in meiosis, we performed a detailed phenotypical analysis of the defects at different stages of

meiosis using homothallic strains with or without the mutation. As shown in Fig 3A, WT *S. pombe* cells usually immediately proceeded to meiosis after nuclear fusion (karyogamy) in which cells with two conjugated nuclei only present transiently. In contrast, in the *akr1* mutant, cells with two closely juxtaposed but not fused nuclei could be readily detected, suggesting that the unfused state was fairly stable. This phenotype was further explored using Nup85, a nucleoporin protein (Chen et al, 2004), tagged with GFP to reveal the

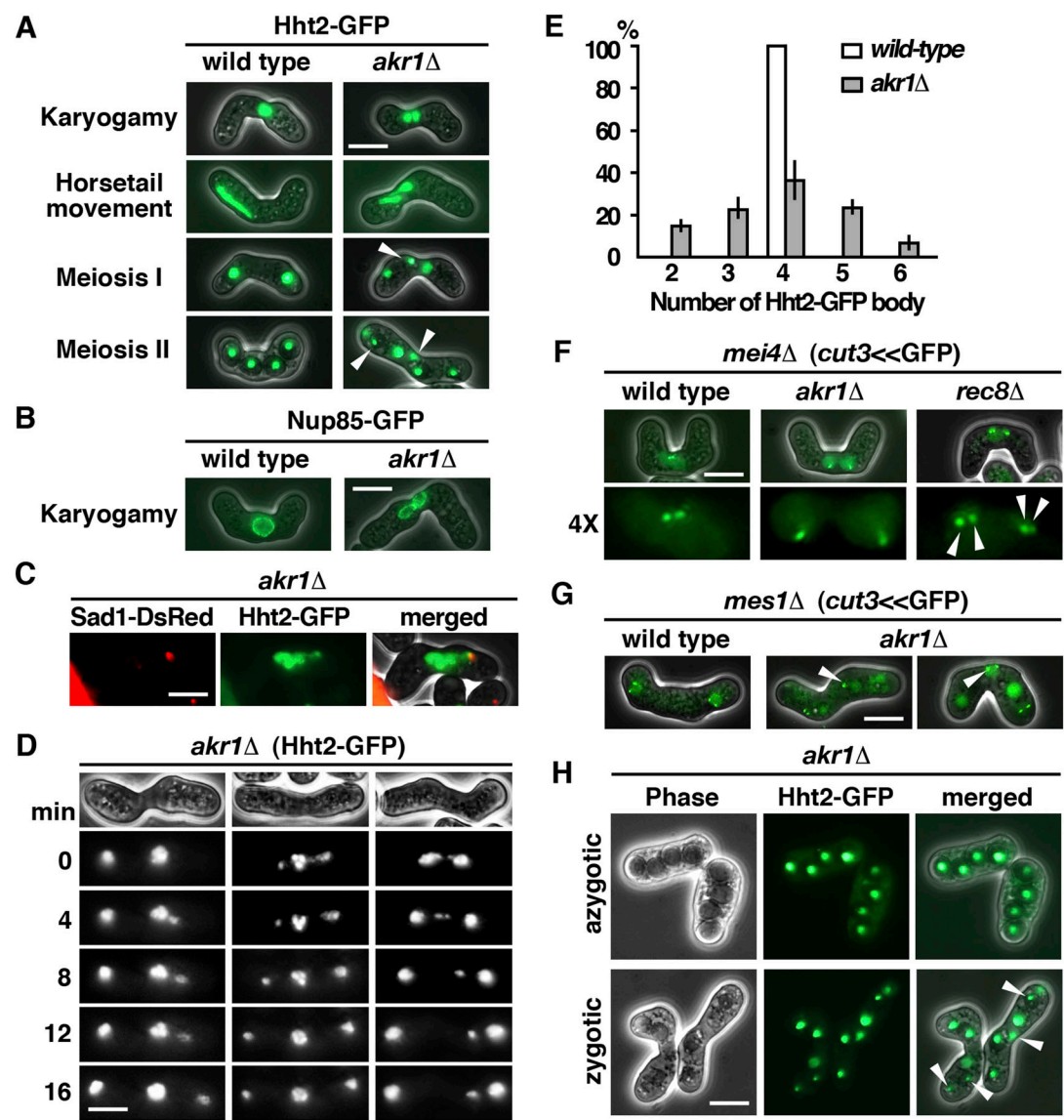

**Figure 3. *akr1* mutant is defective in meiotic nuclear fusion.**
**(A)** Micrographs of meiotic cells of the indicated strains expressing Hht2-GFP protein as the nuclear marker. Scale bar, 5 µm. Arrowheads indicate segregation defects.
**(B)** Micrographs of meiotic cells of the indicated strains expressing Nup85-GFP protein as the nuclear envelop marker. **(C)** Micrographs of the *tht1* cell expressing Hht2-GFP and Sad1-DsRed protein as the spindle pole body marker. **(D)** Time-lapse imaging of Hht2-GFP was performed at 4-min intervals with the *akr1* strain. **(E)** Quantification of the sporulation defect in *akr1* cells. The percentages of all asci containing different numbers of Hht2-GFP nuclear bodies were shown as indicated (error bars represent one SD; the mean of three separate measurements was shown, n = 200). **(F, G)** Micrographs of meiotic cells of the indicated strains arrested at different stages of the meiotic cycle (*mei4* for prophase and *mes1* for meiosis I) expressing *cut3*<<GFP protein as the chromosome marker. Arrowheads indicate aberrant chromosome segregation. **(H)** Micrographs of *akr1* cells undergo zygotic and azygotic meiosis expressing Hht2-GFP protein as the nuclear marker. Arrowheads indicate aberrant spores.

nuclear envelope. As shown in Fig 3B, we found that in the *akr1* mutant, the nuclear envelopes remained largely unfused. Although we could not rule out the possibility that the nuclear membranes might be partially fused at the conjugation site, these results suggested that the *akr1* mutant was defective in nuclear fusion.

After nuclear fusion, prophase was accompanied by the presence of an oscillating horsetail nucleus with the telomeres bundled at the spindle pole body (SPB), generating a bouquet orientation of chromosomes (Chikashige et al, 2006). We found that unlike the single horsetail seen in the WT cells, cells with a pair of small horsetail nuclei lying side by side revealed by Hht2-GFP protein were frequently observed in the *akr1* mutant (Fig 3A). In these twin horsetails, the pointed ends were always on the same side oscillating synchronously, suggesting that these pair of horsetail nuclei were moving in a coordinated fashion. These results implied that after the nuclear conjugation, a single SPB was formed that was able to interact with both haploid nuclei, generating the bouquet structure to lead the horsetail movement, but chromosome homologues were confined in two different compartments during the oscillating movement that failed to interact with each other. In

support of this idea, a single SPB signal revealed by the SUN domain Sad1-DsRed protein (Hagan & Yanagida, 1995) leading the twin horsetails movement was observed in *akr1* mutant (Fig 3C).

At meiosis I, the WT cell diploid nucleus was segregated into two nuclei and generated four equal spores in meiosis II. However, in the *akr1* mutant, chromosomes were frequently not equally segregated into daughters with more than two nuclei at meiosis I (Fig 3D) and generated unequal spores in meiosis II (Fig 3A). As a result, up to 64% of spores bore with more or less than four spores (Fig 3E) containing nuclei in various sizes (Hht2-GFP bodies) with reduced spore viability (23% in the *akr1* mutant compared with 98% in WT cells determined by tetra analysis).

In *S. pombe*, the pairing of homologous chromosomes was achieved during horsetail nuclear movement to establish a physical link for monopolar attachment to ensure accurate chromosome segregation at meiosis I. The chromosome segregation defects of the *akr1* mutant could be a consequence of the defect in nuclear fusion in which chromosome homologues failed to pair and interact for proper chromosome segregation. Alternatively, sister chromatid cohesion could be defective in the *akr1* mutant. To distinguish between these two possibilities, experiments were performed to determine whether *akr1* cells precociously separated their sister chromatid using homothallic strains by monitoring GFP associated with the chromosome arm at the *cut3* locus (*cut3<<GFP*, which is located at the middle of the left arm of chromosome 2) (Win et al, 2006). Cells were arrested in meiotic prophase I by the inactivation of Mei4, a transcription factor required for the progress of the meiotic cycle (Horie et al, 1998). As shown in Fig 3F, two *cut3<<GFP* signals were detected in most of the WT and *akr1* mutant cells, whereas in the meiotic cohesin *rec8* mutant (Watanabe & Nurse, 1999), sister chromosome arms were clearly separated, forming four distinct signals in *mei4*-arrest cells. These results suggested that sister chromatid cohesion was maintained in the *akr1* mutant before entering into meiosis I. However, these cells failed to equally segregate their chromosome into two daughter cells indicated by monitoring *cut3<<GFP* signals arrest at meiosis I by the inactivation of Mes1, a meiotic anaphase-promoting complex (APC) inhibitor (Kimata et al, 2008) (Fig 3G). Consequently, unequal and inaccurate spore formation occurred in the *akr1* mutant. Therefore, the aberrant spore formation was likely produced via the inability to fuse the nuclei resulting in the twin horsetails stage, which was explored further in the following experiments.

In addition to the meiosis after nuclear fusion (zygotic meiosis), meiosis in *S. pombe* can be induced directly from a diploid cell without intervening conjugation (azygotic meiosis). We found that some *akr1* cells went through abnormal sporulation generating dyads in addition to tetrads (Fig 3E), which readily produced diploid cells homozygous for the *akr1* mutation to undergo azygotic meiosis. As shown in Fig 3H, the resulting azygotic asci appeared completely normal with four equal spores. These results suggested that the *akr1* mutation did not interfere with the processes of meiosis and sporulation in azygotic meiosis, which bypassed the requirement of nuclear fusion. Taken together, these results suggested a function of Akr1 in the nuclear fusion that was required for accurate chromosome segregation for proper spore formation.

## S-palmitoylation of Tht1 by Akr1 is required for meiotic nuclear fusion

The meiotic defects of the *akr1* mutant including the unfused nuclei, twin horsetails movement, unequal chromosome segregation, and aberrant spore formation, which were suppressed in azygotic meiosis, resembled those of the *tht1* mutant (Fig 4A–C) (Tange et al, 1998). Tht1 (twin horsetails protein 1) is a meiotic nuclear fusion protein reported but not validated in the SwissPalm—the S-palmitoylation database (Blanc et al, 2015). Experiments were performed to determine whether Tht1 might be the target of Akr1 responsible for the meiotic defects observed. Tht1 was a meiosis-specific protein. To facilitate the characterization, we generated genomic N-terminally tagged GFP fusion Tht1 protein, ectopically expressed in exponentially growing cells by the thiamine-repressible *nmt1* promoter. Similar to Akr1, Tht1 is an ER protein localized at the cell peripheral and nuclear membrane (Fig 4D). In the absence of Akr1, Tht1 was mislocalized from the ER as indicated by granule-like structures observed in the log phase and in meiotic cells. These results suggested a role of Akr1 in regulating Tht1 function, probably mediated by S-palmitoylation.

We further carried out the acyl-resin–assisted capture (acyl-RAC) experiments (Tewari et al, 2020) to determine the S-palmitoylation of Tht1. Neutral hydroxylamine (HA) was used to cleave the thioester bond between palmitate and the modified cysteine residue. The consequently revealed free sulfhydryl group was captured by thioreactive sepharose. As shown in Fig 4E, GFP-Tht1 was specifically found in the +HA elution, suggesting that Tht1 was S-palmitoylated. Furthermore, the level of GFP-Tht1 recovered in *akr1* cells was greatly reduced (27% in *akr1* mutant as compared with WT cells), suggesting it was not efficiently S-palmitoylated in the *akr1* mutant. To determine the site in which Akr1 S-palmitoylated Tht1, a GFP-trap pull-down experiment followed by MALDI MS/MS analysis was performed. As shown in Fig 4F, peptides of Tht1 with 62.5% protein sequence coverage were recovered in the lysate. Our MALDI MS/MS analysis revealed that Tht1 was S-palmitoylated at cysteine 65 and 78. Furthermore, we found that the level of Tht1 mutant protein with cysteine residue 65 and 78 replaced by alanine residue was greatly reduced when moderately expressed from the *nmt41* promoter (Fig 4G and H) and failed to rescue the meiotic defect of *tht1* mutant that was revealed using Htb1 (histone H2B) mono cherry fusion protein as the nuclear marker (Fig 4I). Given that both proteins were ectopically expressed from the same *nmt1* promoter, these results suggested that S-palmitoylation by Akr1 at Cys65 and Cys78 stabilized and localized Tht1 to ER to function in nuclear fusion. In support of these results, a function of protein S-palmitoylation in stabilizing protein has been suggested (Linder & Deschenes, 2007).

## Tht1 interacted with Sey1 to function in karyogamy

In addition to Tht1, peptides of the ER fusion GTPase Sey1 (Rogers et al, 2013) with 44.8% protein sequence coverage were recovered in the lysate of our GFP-trap pull-down experiment (Fig 4F). The high

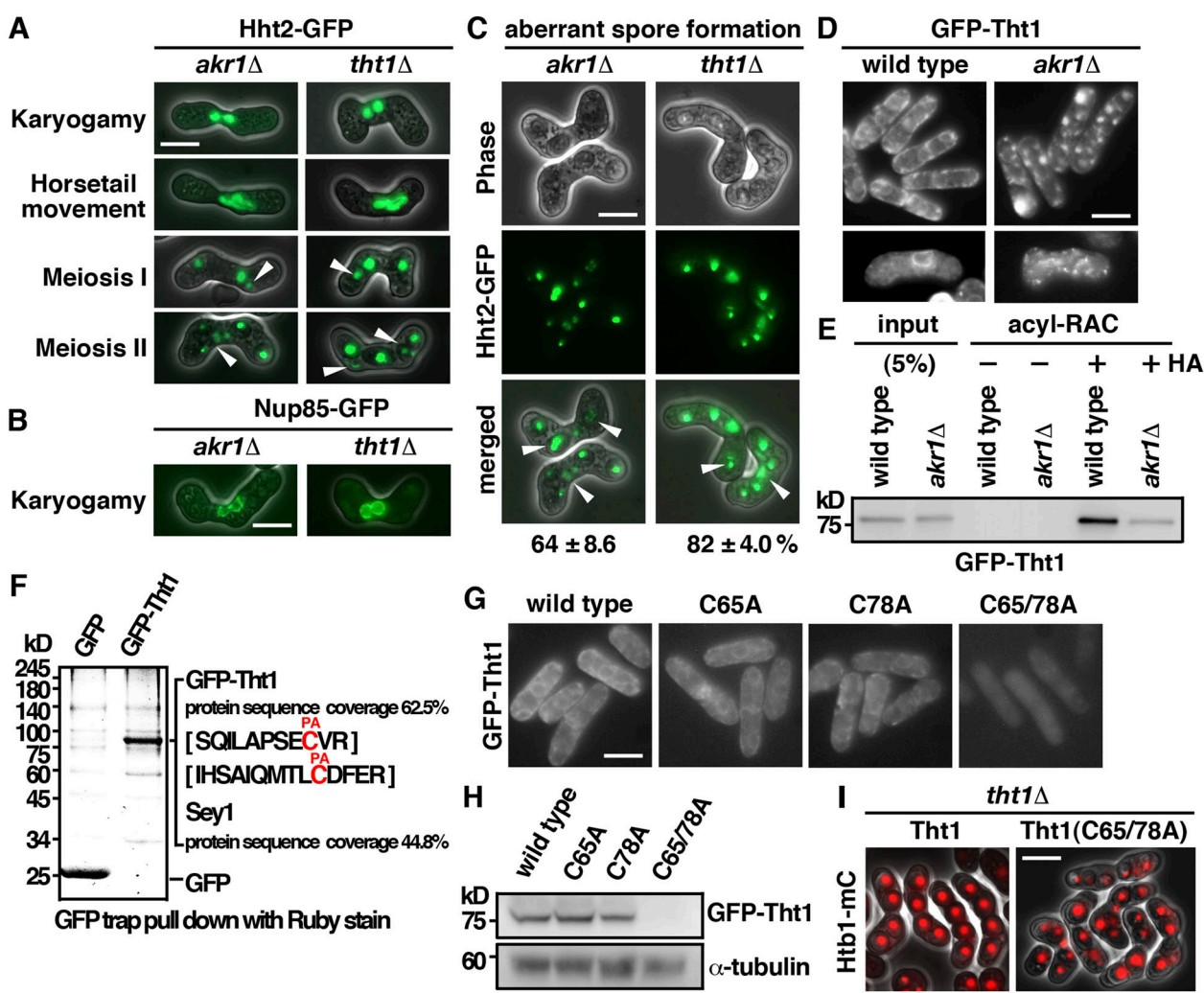

**Figure 4. S-Palmitoylation of Tht1 by Akr1 is required for meiotic nuclear fusion.**
**(A)** Micrographs of meiotic cells of the indicated strains expressing Hht2-GFP protein as the nuclear marker. Scale bar, 5 μm. Arrowheads indicate segregation defects. **(B)** Micrographs of meiotic cells of the indicated strains expressing Nup85-GFP protein as the nuclear envelop marker. **(C)** Quantification of the sporulation defect in *tht1* cells. The percentages of asci with unequal and inaccurate spore formation revealed by the numbers of Hht2-GFP nuclear bodies were expressed as means ± SD in triplicate. (n = 200). **(D)** Micrographs of log-phase and meiotic cells of the indicated strains expressing GFP-Tht1 protein. **(E)** Samples from acyl-RAC experiments were subjected to immunoblotting using an anti-GFP antibody to reveal proteins of interest. Specifically, purified palmitoyl-proteins can be seen enriched in the +HA samples. 10% of the input was loaded onto the gel. **(F)** GFP-Trap pulldown of GFP-Tht1 proteins resolved by SDS–PAGE was visualized by SYPRO Ruby staining. The identities of the constituent proteins and S-palmitoylated peptides identified by MALDI-MS/MS analysis are indicated on the right. **(G)** Micrographs of cells expressing WT or mutant GFP-Tht1 proteins. **(H)** Whole-cell protein extracts prepared from cells expressing WT or mutant GFP-Tht1 proteins were separated by SDS–PAGE and subjected to immunoblotting using an anti-GFP antibody to reveal proteins of interest. Antibody against α-tubulin was used as the loading control. **(I)** Micrographs of homothallic *tht1* cells transformed with a plasmid expressing WT Tht1 or C65/78A mutant. Htb1-mCherry proteins were used as the nuclear marker.

coverage rate suggested that Sey1 was a potential interacting protein of Tht1. In support of this idea, we found that Sey1 specifically interacted with Tht1 but not with the GFP control as demonstrated by coimmunoprecipitation (Fig 5A). Furthermore, meiotic defects similar to that of the *tht1* mutant including the twin horsetails movement, unfused nuclei, unequal chromosome segregation, and aberrant spore formation were observed in the *sey1* mutant (Fig 5B–D). However, the frequency of defective meiotic cells was much lower in the *sey1* mutant (38% as compared with 82% in the *tht1* mutant), suggesting that in addition to Sey1, Tht1 required other proteins to facilitate chromosome segregation. Nevertheless, the similar meiotic defects suggested a role of Tht1-Sey1 in

reorganizing ER to facilitate nuclear fusion for faithful meiotic chromosome segregation.

## Discussion

Although changes in the global protein S-palmitoylation pattern have been associated with different physiological states such as cellular differentiation (Zhang & Hang, 2017) and pathological conditions (Zhang et al, 2021), the detailed molecular mechanisms of how protein S-palmitoylation regulates these processes remained to be defined.

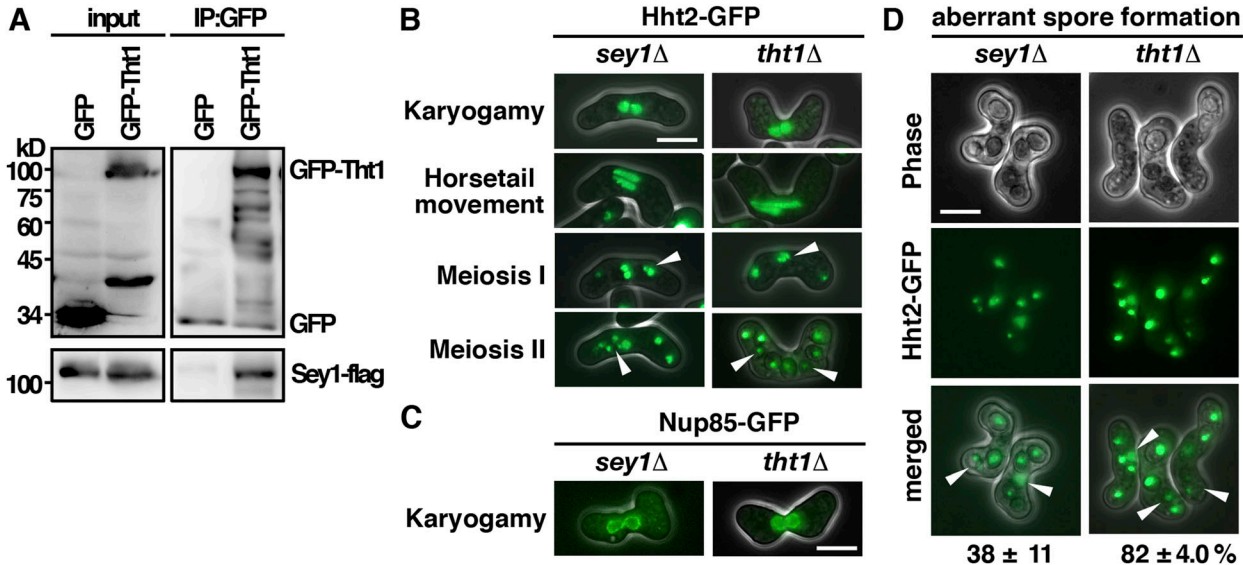

**Figure 5. Tht1 interacted with Sey1 to function in the nuclear function.**
**(A)** Coimmunoprecipitation was performed with extracts prepared from Sey1-flag–tagged strains expressing GFP-Tht1 or GFP control proteins. GFP-Trap affinity resin was used to pull down GFP proteins. Immunoprecipitation was then analyzed by Western immunoblotting with antibodies against GFP and flag. Five percent of the input was loaded onto the gel. **(B)** Micrographs of meiotic cells of the indicated strains expressing Hht2-GFP protein as the nuclear marker. Scale bar, 5 $\mu$m. Arrowheads indicate segregation defects. **(C)** Micrographs of meiotic cells of the indicated strains expressing Nup85-GFP protein as the nuclear envelop marker. **(D)** Quantification of the sporulation defect in *sey1* cells. The percentages of asci with unequal and inaccurate spore formation revealed by the numbers of Hht2-GFP nuclear bodies were expressed as means ± SD in triplicate. (n = 200).

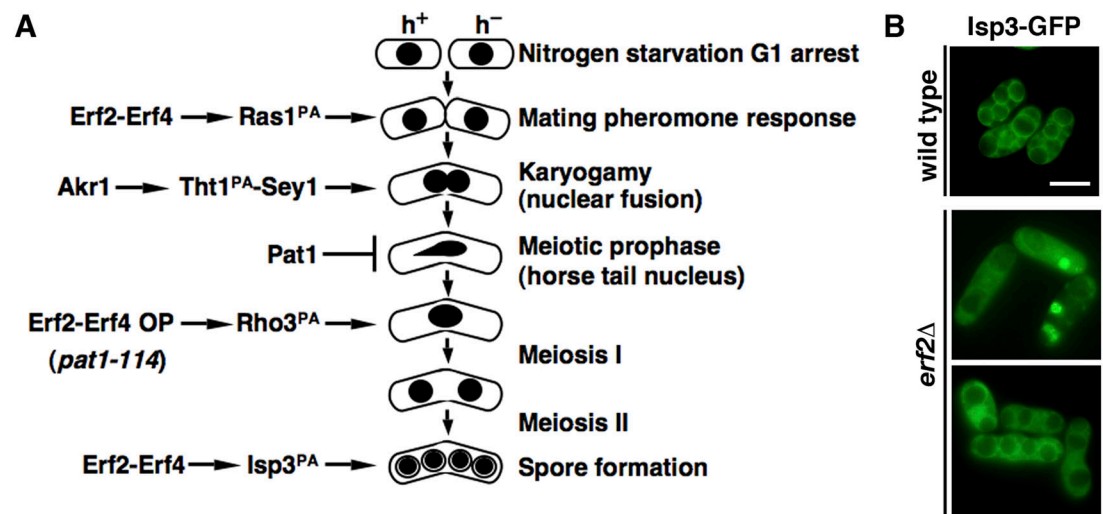

**Figure 6. S-palmitoylation is required at multiple stages of meiosis mediated by Erf2-Erf4 and Akr1 palmitoylacyltransferases in *Schizosaccharomyces pombe*.**
**(A)** Schematic representation of the involvements of S-palmitoylation at different stages of meiosis. **(B)** Micrographs of *pat1*-induced haploid meiotic cells of the indicated strains expressing Isp3-GFP. Scale bar, 5 $\mu$m.

In an earlier study of protein S-palmitoylation in *S. pombe*, using the synchronized meiosis induced from *pat1-114* diploid cells, Zhang et al (2013) demonstrated that the global protein S-palmitoylation pattern was significantly altered and shaped by varying Erf2 palmitoylacyltransferase activity during meiosis. Meiotic entry was delayed in the *erf2* mutant and increasing Erf2 palmitoylacyltransferase activity trigged aberrant meiosis in sensitized cells. Multiple connate substrates of Erf2 were identified. A function of

Rho3 S-palmitoylated by Erf2 in the meiotic entry was suggested. However, this work, performed primarily in azygotic diploid cells, largely overlooked the early processes of cellular differentiation such as zygote formation (Fig 6A), and no detailed molecular mechanism was suggested.

Here, we tracked cells throughout the natural sexual life cycle and performed a detailed phenotypic analysis of these palmitoylacyltransferases mutants. We found that Ras1, but not Rho3, was

the primary target of Erf2-Erf4 in meiosis. Similar to the *ras1* mutant, mutants of *erf2* and *erf4* were sterile and failed to form any zygote (Fig 2A). Compartment-specific signaling by Ras1 has been reported (Chang et al, 1994; Onken et al, 2006). We further demonstrated that in the absence of Erf2-Erf4, Ras1 was mislocalized in the cytoplasm and failed to enrich at the cell conjugation site for mating pheromone response (Fig 2B). Thus, S-palmitoylation of Ras1 by Erf2-Erf4 is intimately involved in controlling mating in *S. pombe*.

In addition, a function of Akr1 in karyogamy was demonstrated. The nuclear fusion protein Tht1 was identified as the target of Akr1 responsible for the meiotic phenotype. S-palmitoylation is required for Tht1 protein stability and cellular localization to ER to function for nuclear fusion. Furthermore, the ER fusion GTPase Sey1 was identified as an interacting protein of Tht1 (Fig 5A). A role of Tht1-Sey1 in reorganizing ER to facilitate nuclear fusion for faithful meiotic chromosome segregation was suggested. Experiments are in progress to test this hypothesis.

The results described above established the vital roles of protein S-palmitoylation at the early stage of meiosis (Fig 6A). The spore coat protein Isp3 is another substrate of Erf2-Erf4 (Zhang et al, 2013). Using *pat1*-induced haploid meiosis to overcome the *erf2* mutation–induced block in meiotic entry, we found that in the absence of Erf2, Isp3 was mislocalized from the spore coat (Fig 6B). These results suggested a role of protein S-palmitoylation in regulating Isp3 function at the later stage of meiosis. Taken together, our study demonstrated that protein S-palmitoylation is required at both early and late stages of meiosis to ensure proper cellular differentiation (Fig 6A). Furthermore, using comparative proteomics, 238 proteins were suggested to be preferentially S-palmitoylated in meiosis by Erf2, the major meiotic palmitoylacyltransferase in *S. pombe* (Zhang et al, 2013). These proteins are involved in diverse cellular processes (Fig S3A). Among them, mutants of 38% proteins (56 out of 146 characterized) were identified with meiotic phenotypes (mating, chromosome segregation, or sporulation defects) in two genome-wide screens (Fig S3B and C) (Dudin et al, 2017; Blyth et al, 2018). Characterizing the role of protein S-palmitoylation in the meiotic function of these proteins will provide further insight into our understanding of the fundamental mechanism coordinating cell differentiation in general. However, even without addressing this point, the work alone and analysis described here highlight the vital role of protein S-palmitoylation in meiosis at different stages with the versatility of the complicated regulatory network involved, which has significantly advanced our understanding of the biological function of protein S-palmitoylation in eukaryotes.

# Materials and Methods

### Fission yeast strains and methods

We constructed all strains according to standard procedures. Information of oligonucleotides for gene disruption or modification can be obtained upon request. The original GFP-*ras1* (Myo52-tdTomato), GFP-*rho3*, and histone GFP, mono cherry–tagged strains were gifts from Sophie G Martin, Reiko Sugiura,

and Yasushi Hiraoka. A complete list of the strains used in this study is given in Table 1. Conditions for growth, maintenance, and genetic manipulation of fission yeast were as described previously (Hsiao et al, 2020). Except otherwise stated, strains were grown at 30°C in yeast extract or Edinburgh Minimal Medium (EMM2) with appropriate supplements. Where necessary, gene expression from the *nmt1* promoter was repressed by the addition of 15 mM thiamine to the culture medium. Homothallic strains were spotted onto a nitrogen-free medium to induce meiosis at room temperature.

### Plasmid construction

Plasmids for the expression of GFP-Ras1 and GFP-Tht1 (1,549–1,751) were constructed by PCR amplification of the corresponding ORF of cDNA derived from the genomic N-terminal GFP–tagged *S. pombe* strains and subsequently ligated into plasmids derived from pREP1 at the multiple cloning sites. The Ras1 (C215A) and Tht1 (C65A and C78A) mutants were created using PCR-based site-directed mutagenesis and verified by DNA sequencing.

### Microscopy

Visualization of GFP-, mono cherry–, and DsRed-tagged proteins in living cells was performed at room temperature. Images acquired from a Leica DM RA2 microscope equipped with a Leica DC 350F camera were assembled using Adobe Photoshop. Visualization of Hht2-GFP in living cells embedded in 0.6% LMP agarose was performed at room temperature as previously described (Win et al, 2006).

### Antibodies and immunoprecipitation

The antibody against GFP (ab290) was purchased from Abcam. The anti-Flag M2 antibody was from Sigma-Aldrich. Antibody against $\alpha$-tubulin (Sigma-Aldrich) was used as a control. For immunoprecipitation from yeast extracts, $2 \times 10^8$ yeast cells were lysed in 200 ml NP-40 lysis buffer (6 mM $Na_2HPO_4$, 4 mM $NaH_2PO_4$, 1% NP-40, 150 mM NaCl, 2 mM EDTA, 50 mM NaF, 0.1 mM $Na_3VO_4$, 1 mM PMSF, 1 mM DTT, complete protease inhibitor cocktail) by vortexing with acid-washed glass beads. The lysate was clarified by centrifugation, and GFP fusion proteins were retrieved using GFP-Trap-coupled agarose beads (ChromoTek) following the manufacturer's instructions.

### GFP protein identification

The purified proteins were separated on SDS–PAGE gels and visualized by Sypro Ruby (Invitrogen). Protein bands were excised from gels, treated with trypsin, and subjected to analysis for protein identification using the Waters NanoAcquity UPLC-Synapt G2 HDM instrument by the Core Facilities for Proteomics at National Health Research Institutes. Data were processed for database searching using ProteinLynx Global Server against the Swiss-Prot database with the Mascot program.

**Table 1.** *Schizosaccharomyces pombe* strains used in this study.

| Genotype | Source |
|---|---|
| *h⁻pREP1-akr1-GFP leu1-32* | This study |
| *h⁻ pREP1-erf2-GFP leu1-32* | This study |
| *h⁻ pREP1-erf4-GFP leu1-32* | This study |
| *h⁻ pREP1-pfa3-GFP leu1-32* | This study |
| *h⁻ pREP1-pfa5-GFP leu1-32* | This study |
| *h⁻ pREP1-swf1-GFP leu1-32* | This study |
| *h⁹⁰ hht2-GFP::ura4 leu1-32 ura4-D18* | Yasushi Hiraoka |
| *h⁹⁰ hht2-GFP::ura4 akr1::hyh* (hygromycin B phospho-transferase) *ura4-D18* | This study |
| *h⁹⁰ hht2-GFP::ura4 erf2::hyh ura4-D18* | This study |
| *h⁹⁰ hht2-GFP::ura4 erf4::hyh ura4-D18* | This study |
| *h⁹⁰ hht2-GFP::ura4 pfa3::hyh ura4-D18* | This study |
| *h⁹⁰ hht2-GFP::ura4 pfa5::hyh ura4-D18* | This study |
| *h⁹⁰ hht2-GFP::ura4 ras1::hyh ura4-D18* | This study |
| *h⁹⁰ hht2-GFP::ura4 rho3::hyh ura4-D18* | This study |
| *h⁹⁰ GFP-ras1 myo52-tdTomato::natMX* | Sophie G Martin |
| *h⁹⁰ GFP-ras1 myo52-tdTomato::natMX erf2::hyh* | This study |
| *h⁹⁰ GFP-rho3::LEU2 myo52-tdTomato::natMX leu1-32* | Reiko Sugiura for *GFP-rho3* |
| *h⁹⁰ GFP-rho3::LEU2 myo52-tdTomato::natMX erf2::hyh leu1-32* | This study |
| *h⁹⁰ htb1-mCherry:: natMX pREP1-GFP-ras1 leu1-32* | Yasushi Hiraoka for *htb1-mCherry* |
| *h⁹⁰ htb1-mCherry::natMX pREP1-GFP-ras1(C215A) leu1-32* | This study |
| *h⁹⁰ nup85-GFP::kanʳ* | This study |
| *h⁹⁰ nup85-GFP::kanʳ akr1::hyh* | This study |
| *h⁹⁰ hht2-GFP::ura4 sad1-DsRed::LEU2 akr1::kanʳ leu1-32 ura4-D18* | This study |
| *h⁹⁰ mei4::ura4 cut3<<GFP (cut3::lacOp his7::lacI-GFP) ura4-D18* | Lab stock |
| *h⁹⁰ mei4::ura4 cut3<<GFP akr1::hyh ura4-D18* | This study |
| *h⁹⁰ mei4::ura4 cut3<<GFP rec8:: kanʳ ura4-D18* | Lab stock |
| *h⁹⁰ mes1::LEU2 cut3<<GFP leu1-32* | This study |
| *h⁹⁰ mes1::LEU2 cut3<<GFP akr1::hyh leu1-32* | This study |
| *h⁹⁰ hht2-GFP::ura4 tht1::hyh ura4-D18* | This study |
| *h⁹⁰ nup85-GFP::kanʳ tht1::hyh* | This study |
| *h⁹⁰ p3-GFP-tht1::kanʳ* | This study |
| *h⁹⁰ p3-GFP-tht1:: kanʳ akr1::hyh* | This study |
| *h⁻pREP41-GFP-tht1 leu1-32* | This study |
| *h⁻pREP41-GFP-tht1(C65A) leu1-32* | This study |
| *h⁻pREP41-GFP-tht1(C78A) leu1-32* | This study |
| *h⁻pREP41-GFP-tht1(C65/78A) leu1-32* | This study |
| *h⁹⁰ htb1-mCherry:: natMX pREP1-tht1 leu1-32* | This study |
| *h⁹⁰ htb1-mCherry:: natMX pREP1-tht1(C65/78A) leu1-32* | This study |
| *h⁻ p3-GFP-tht1:: kanʳ sey1-flag:: kanʳ* | This study |
| *h⁹⁰ hht2-GFP::ura4 sey1:: hyh ura4-D18* | This study |
| *h⁹⁰ nup85-GFP:: kanʳ sey1::hyh* | This study |
| *h⁻ pat1-114 isp3-GFP:: kanʳ* | This study |
| *h⁻ pat1-114 isp3-GFP:: kanʳ erf2::hyh* | This study |

## Acyl-RAC assay

The palmitoylation assay was conducted based on the acyl-RAC assay described previously (Tewari et al, 2020) with the following modifications. Cell lysate extracted from $3 \times 10^9$ *S. pombe* cells in ice-cold lysis buffer (PBS, pH 7.4, 0.5 mM EDTA, protease inhibitor [PI], and 1 mM phenylmethylsulfonyl fluoride) was incubated with 1% Triton X-100 and 25 mM N-ethylmaleimide (NEM) for 1 h. The protein pellet was collected by chloroform-methanol (CM) precipitation, which may be stored at −80°C for up to 2 wk for the following assay.

For acyl-RAC, the protein pellet was dissolved in a small volume of 4% SDS buffer (4% SDS, 50 mM Tris–HCl, pH 7.4, 5 mM EDTA) incubated at 37°C to improve dissolution before free cysteines blocking with 20 mM N-ethylmaleimide (NEM) in LB buffer (150 mM NaCl, 50 mM Tris–HCl, pH 7.4, 5 mM EDTA) overnight. Three sequential CM precipitations were processed to remove excessive NEM, and the sample was resolved in LB buffer with 0.2% SDS before treatment with neutral 1 M hydroxylamine (HAM) to cleave the thioester bound between cysteine residues and a fatty acid moiety. 30 μl of thiopropyl resin (G-Biosciences) was added to every tube and incubated for 1–2 h at RT. The beads were washed five times with washing buffer (LB buffer with 0.1% Triton X-100) to remove unbound proteins, and the palmitoylated proteins were eluded in a SDS sample buffer with DTT detected by Western blotting.

# Supplementary Information

# Acknowledgements

We thank Y Herry Sun for funding support; Sophie G Martin, Reiko Sugiura, and Yasushi Hiraoka for yeast strains; and Mingzi M Zhang for critical review of the manuscript. This work was supported by the National Health Research Institute (MG-110-PP-07). Taiwan. T-V Pham was supported by the NHRI-NCU Graduate Student Program.

## Author Contributions

T-V Pham: investigation and writing—original draft, review, and editing.
W-Y Hsiao: investigation.
Y-T Wang: investigation.
S-D Yeh: supervision and writing—review and editing.
S-W Wang: conceptualization, investigation, and writing—original draft, review, and editing.

## Conflict of Interest Statement

The authors declare that they have no conflict of interest.

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
