## [Reviewer comments · Life Science Alliance]

Life Science Alliance

Protein S-palmitoylation regulates different stages of meiosis in *Schizosaccharomyces pombe*

Thanh-Vy Pham, Wan-Yi Hsiao, Yi-Ting Wang, Shu-Dan Yeh, and Shao-Win Wang

DOI: <https://doi.org/10.26508/lsa.202201755>

Corresponding author(s): Shao-Win Wang, National Health Research Institutes

Review Timeline:

Submission Date:	2022-10-06
Editorial Decision:	2022-10-31
Revision Received:	2022-12-14
Editorial Decision:	2023-01-03
Revision Received:	2023-01-06
Accepted:	2023-01-06

Scientific Editor: Novella Guidi

Transaction Report:

October 31, 2022

Re: Life Science Alliance manuscript #LSA-2022-01755-T

Dr. Shao-Win Wang
National Health Research Institutes
Institute of Molecular & Genomic Medicine
35 Keyan Road
Zhunan 35053
Taiwan

Dear Dr. Wang,

Thank you for submitting your manuscript entitled "Protein S-palmitoylation regulates different stages of meiosis in *Schizosaccharomyces pombe*" to Life Science Alliance. The manuscript was assessed by expert reviewers, whose comments are appended to this letter. We invite you to submit a revised manuscript addressing the Reviewer comments.

Thank you for this interesting contribution to Life Science Alliance. We are looking forward to receiving your revised manuscript.

Sincerely,

B. MANUSCRIPT ORGANIZATION AND FORMATTING:

Reviewer #1 (Comments to the Authors (Required)):

Comments to the Authors:

The evaluated manuscript documents the role of protein S-palmitoylation in the regulation of meiosis in fission yeast. While the function of Erf2 and Erf4 proteins has already been studied <https://doi.org/10.1371/journal.pbio.1001597>. The role of Akr1 in karyogamy during meiosis of fission yeast is novel. The authors follow different approaches to determine the interaction of protein Tht1 with Sey1. They suggest the role of Sey1 in karyogamy as well. There are, however, a number of issues and concerns listed below that should be addressed to improve or clarify some points in the manuscript.

Major points

#1

On page 5, line 8 is written, that all studied S-palmitoylacyltransferases were ectopically overexpressed from a strong nmt1 promoter. The overexpression can lead to ectopic localization. Using integrative vectors with native/milder promoters would reflect a more natural localization of proteins.

The localization experiments in Figure 1 would be strengthened considerably if the authors could demonstrate the colocalization of different S-palmitoylacyltransferases with marker proteins from stated (ER, Golgi) cellular compartments.

#2 On page 8, Pham et al. state, that a single SPB is formed, but data from Figure 3A do not support the results. SPB labeling could confirm it. The authors calculated the number of GFP dots after meiosis, however, the sporulation defect is difficult to calculate from Figure 3A. For better characterization of akr1 mutation, live-cell imaging should be used. In addition, spore viability would help to understand the penetrance of the phenotype better.

#3 Could you specify in material and methods the production of akr1 diploid cells used for azygotic meiosis in Figure 3E?

#4 Page 11, line 8. The authors state that mutation of Tht1 C65/78A leads to reduce expression of the Tht1 protein. It would be also useful to show whether meiosis is impaired in single or double mutants.

Minor

#1 Information about the genetic background of the strains is often incomplete. Are all auxotrophic mutations written? What does it mean hyh?

#2 The spelling, mainly S-palmitoylacyltransferase and horsetail should be checked in the text.

Reviewer #2 (Comments to the Authors (Required)):

It is well established that S-palmitoylation of proteins is important for their cellular localisations and signalling. In this manuscript, Pham et al. characterised two fission yeast palmitoyacyltransferases, Erf2 and Akr1, in particular, focussing on their roles in the meiotic pathway. They showed that Erf2, together with its binding partner Erf4, is required for conjugation; their mutants become sterile. They identified Ras1 as a major substrate for Erf2-Erf4. Akr1 is also important for meiosis, but unlike Erf2, the main defect of akr1 mutants lies in the nuclear fusion process/karyogamy. Subsequent experiments uncovered that a nuclear fusion protein Tht1 is a main substrate for Akr1: the authors showed that C65 and C78 residues within Tht1 are the sites of S-palmitoylation by Akr1.

This is a concise, well-executed study of high quality. The experiments are nicely performed, the results are clearly presented and interpretations seem proper. I believe that this study would appeal a wide range of researchers involved in molecular cell biology. I have no major criticisms or concerns, but several points are listed as follows.

Minor points

1) protein localisation study (Figure 1B and page 5)

First of all, do the authors know functionality of GFP-tagged proteins used in this study? Second, how do they know Akr1 is localised to Golgi (ER localisation seems convincing, though)? Third, it is not clear to this referee that Erf2-GFP and Erf4-GFP are localised to the cell periphery. Perhaps, enlarged images showing areas of the cell periphery would be helpful.

2) erf2 and erf4 mutants (Figure 1C)

If Erf2 and Erf4 are required for Ras1 function, these mutants would show morphological defects, as Ras1 is required for cell shape control during vegetative growth phase in addition to mating upon meiotic entry. Is this the case? If Erf2-Erf4 is only functioning during meiosis, these mutants would display normal cell morphology. Please clarify this issue.

3) Tht1-C65A/C78A (page 11 and Figure 4G and 4H)

Do the authors imply that the Tht1-C65A/C78A 4 protein becomes destabilised, by which it could not be detected in the cell or on the SDS-PAGE gel? Or are protein expression levels per se somehow decreased? If the former scenario were likely, are there any previous examples known in which the defect in protein palmitoylation renders this protein less stable?

4) Figure 6B, 6C and 6D

These data seemed to have already been published by other groups. Therefore, they should be shown as supplementary, not as main figures.

Miscellaneous

a) remain to be defined (page 4)
remains to be defined

b) References

Please indicate citations in the following position: "a bouquet orientation of chromosomes" (page 8), "the meiotic cycle" (page 9) and "inactivation of Mes1, a meiotic APC inhibitor" (page 9). Also, please define "APC" (page 9).

c) Taking together (several places)

Taken together

d) Table 1

Please indicate sources/derivations of individual strains listed.

Reviewer #3 (Comments to the Authors (Required)):

The manuscript uses fission yeast as a model system for the study of the role of palmitoylation. Three palmitoyltransferases are shown to be important for sexual differentiation (mating, nuclear fusion and meiosis) and putative substrates identified. Overall, the paper is interesting and of good quality.

Major comments:

Figure 1: The authors look at the localisation of 6 potentially interesting by overexpressing fusion proteins from a very strong promoter, as they consider that the endogenous protein levels may be too low. This is of course not ideal (overexpressed protein localisation is prone to artifacts). Moreover, overexpression may be unnecessary for the three proteins characterised in detail. Akr1 is expressed at relatively high levels (see PomBase) in vegetative cells. Erf2 and Erf4 are clearly meiotic (see Baehler website, accessible from PomBase). I think it is essential that these 3 proteins are tagged with GFP in their endogenous locus.

Minor comments

1. See also comment above. Discuss the caveats that the proteins are strongly overexpressed.
2. Remove the 3D effects of the cartoons of figure 1a, they just reduce visibility.
3. Chang et al. PMID:7923372 already described the mating phenotype of ras1 mutants, they should be cited.
4. Many references in the reference list are incomplete.
5. Figure 6 is referred to only at the discussion, even though it contains experimental data. I suggest that the discussion and the results are fused

Dear Editor:

Please find attached our revised manuscript. We thank the referees for their constructive comments that help to improve the manuscript. We have thoroughly addressed all these comments and accommodated their suggestion.

A point-by-point response to the issues raised is given below.

Reviewer #1 (Comments to the Authors (Required)):

Comments to the Authors:

The evaluated manuscript documents the role of protein S-palmitoylation in the regulation of meiosis in fission yeast. While the function of Erf2 and Erf4 proteins has already been studied <https://doi.org/10.1371/journal.pbio.1001597>. The role of Akr1 in karyogamy during meiosis of fission yeast is novel. The authors follow different approaches to determine the interaction of protein Tht1 with Sey1. They suggest the role of Sey1 in karyogamy as well. There are, however, a number of issues and concerns listed below that should be addressed to improve or clarify some points in the manuscript.

Major points

#1 On page 5, line 8 is written, that all studied S-palmitoylacyltransferases were ectopically overexpressed from a strong *nmt1* promoter. The overexpression can lead to ectopic localization. Using integrative vectors with native/milder promoters would reflect a more natural localization of proteins.

Our response: As suggested, experiments of Fig 1B were performed with milder expression condition.

The localization experiments in Figure 1 would be strengthened considerably if the authors could demonstrate the colocalization of different S-palmitoylacyltransferases with marker proteins from stated (ER, Golgi) cellular compartments.

Our response: ER red tracker and BODIPY ceramide for Golgi structure were applied to the studies (Fig 1C and D).

#2 On page 8, Pham et al. state, that a single SPB is formed, but data from Figure 3A do not support the results. SPB labeling could confirm it.

Our response: *Sad1-DsRed* was used as the marker of SPB (Fig 3C).

The authors calculated the number of GFP dots after meiosis, however, the sporulation defect is difficult to calculate from Figure 3A. For better characterization of *akr1* mutation, live-cell imaging should be used.

Our response: Time-lapse imaging of Hht2-GFP with the *akr1* strain was provided (Fig 3D).

In addition, spore viability would help to understand the penetrance of the phenotype better.

Our response: Spore viability was provided (Page 9).

#3 Could you specify in material and methods the production of *akr1* diploid cells used for azygotic meiosis in Figure 3E?

Our response: We found that some *akr1* Δ cells went through abnormal sporulation generating dyads in addition to tetrads (Fig. 3E), which readily produced diploid cells homozygous for the *akr1* mutation to undergo azygotic meiosis. As shown in Fig 3H, the resulting azygotic asci appeared completely normal with four equal spores (Page 10).

#4 Page 11, line 8. The authors state that mutation of Tht1 C65/78A leads to reduce expression of the Tht1 protein. It would be also useful to show whether meiosis is impaired in single or double mutants.

Our response: Furthermore, we found that the level of Tht1 mutant protein with cysteine residue 65 and 78 replaced by alanine residue was greatly reduced when moderately expressed from the *nmt41* promoter (Figs 4G and H), and failed to rescue the meiotic defect of *tht1* mutant revealed using Htb1 (histone H2B) mono cherry-fusion protein as the nuclear marker (Fig 4I).

Minor

#1 Information about the genetic background of the strains is often incomplete. Are all auxotrophic mutations written? What does it mean *hyh*?

Our response: Information was added to the strain list. Resistance to antibiotic hygromycin is conferred by the *hyh* gene encoding a hygromycin B phosphotransferase isolate from an *E. coli* strain.

#2 The spelling, mainly S-palmitoylacyltransferase and horsetail should be checked in the text.

Our response: S-palmitoylacyltransferase and horsetail spelling checked.

Reviewer #2 (Comments to the Authors (Required)):

It is well established that S-palmitoylation of proteins is important for their cellular localisations and signalling. In this manuscript, Pham et al. characterised two fission yeast palmitoyacyltransferases, Erf2 and Akr1, in particular, focussing on their roles in the meiotic pathway. They showed that Erf2, together with its binding partner Erf4, is required for conjugation; their mutants become sterile. They identified Ras1 as a major substrate for Erf2-Erf4. Akr1 is also important for meiosis, but unlike Erf2, the main defect of *akr1* mutants lies in the nuclear fusion process/karyogamy. Subsequent experiments uncovered that a nuclear fusion protein Tht1 is a main substrate for Akr1: the authors showed that C65 and C78 residues within Tht1 are the sites of S-palmitoylation by Akr1.

This is a concise, well-executed study of high quality. The experiments are nicely performed, the results are clearly presented and interpretations seem proper. I believe that this study

would appeal a wide range of researchers involved in molecular cell biology. I have no major criticisms or concerns, but several points are listed as follows.

Minor points

1) protein localisation study (Figure 1B and page 5)

First of all, do the authors know functionality of GFP-tagged proteins used in this study?

Our response: Except for Erf2-Erf4, the functionalities of these proteins in *S. pombe* have not been described. Experiments are under way to identify their substrates to determine the functionality of these proteins (Page 6).

Second, how do they know Akr1 is localised to Golgi (ER localisation seems convincing, though)?

Our response: ER red tracker and BODIPY ceramide for Golgi structure were applied to the studies (Fig 1C and D).

Third, it is not clear to this referee that Erf2-GFP and Erf4-GFP are localised to the cell periphery. Perhaps, enlarged images showing areas of the cell periphery would be helpful.

Our response: Enlarged images were provided (Fig 1B).

2) erf2 and erf4 mutants (Figure 1C)

If Erf2 and Erf4 are required for Ras1 function, these mutants would show morphological defects, as Ras1 is required for cell shape control during vegetative growth phase in addition to mating upon meiotic entry. Is this the case? If Erf2-Erf4 is only functioning during meiosis, these mutants would display normal cell morphology. Please clarify this issue.

Our response: In addition to mating upon meiotic entry, Ras1 is also required for cell shape control during the vegetative growth phase. However, we found that, in contrast to the pear-shaped vegetative *ras1* cells, *erf2* and *erf4* mutants displayed relatively normal cell morphology (Fig 2E). These results suggested that the requirement of Erf2-Erf4 for Ras1 function seems only during meiosis (Page 8).

3) Tht1-C65A/C78A (page 11 and Figure 4G and 4H)

Do the authors imply that the Tht1-C65A/C78A protein becomes destabilised, by which it could not be detected in the cell or on the SDS-PAGE gel? Or are protein expression levels per se somehow decreased? If the former scenario were likely, are there any previous examples known in which the defect in protein palmitoylation renders this protein less stable?

Our response: Furthermore, we found that the level of Tht1 mutant protein with cysteine residue 65 and 78 replaced by alanine residue was greatly reduced when moderately expressed from the *nmt41* promoter (Figs 4G and H), and failed to rescue the meiotic defect of *tht1* mutant revealed using Htb1 (histone H2B) mono cherry-fusion protein as the nuclear marker (Fig 4I).

4) Figure 6B, 6C and 6D

These data seemed to have already been published by other groups. Therefore, they should be

shown as supplementary, not as main figures.

Our response: As suggested, Figs 6B, 6C, and 6D moved to supplementary.

Miscellaneous

a) remain to be defined (page 4)
remains to be defined

Our response: Spelling checked.

b) References

Please indicate citations in the following position: "a bouquet orientation of chromosomes" (page 8), "the meiotic cycle" (page 9) and "inactivation of Mes1, a meiotic APC inhibitor" (page 9). Also, please define "APC" (page 9).

Our response: References added.

c) Taking together (several places)
Taken together

Our response: Spelling checked.

d) Table 1

Please indicate sources/derivations of individual strains listed.

Our response: Information was added to the strain list.

Reviewer #3 (Comments to the Authors (Required)):

The manuscript uses fission yeast as a model system for the study of the role of palmitoylation. Three palmitoyltransferases are shown to be important for sexual differentiation (mating, nuclear fusion and meiosis) and putative substrates identified. Overall, the paper is interesting and of good quality.

Major comments:

Figure 1: The authors look at the localisation of 6 potentially interesting by overexpressing fusion proteins from a very strong promoter, as they consider that the endogenous protein levels may be too low. This is of course not ideal (overexpressed protein localisation is prone to artifacts). Moreover, overexpression may be unnecessary for the three proteins characterised in detail. Akr1 is expressed at relatively high levels (see PomBase) in vegetative cells. Erf2 and Erf4 are clearly meiotic (see Baehler website, accessible from PomBase). I think it is essential that these 3 proteins are tagged with GFP in their endogenous locus.

Our response: As suggested by Reviewer #1, experiments of Fig 1B were performed with milder expression condition.

Minor comments

1. See also comment above. Discuss the caveats that the proteins are strongly overexpressed.

Our response: The localizations of these proteins were essentially identical to those when cultured in the absence of thiamine expressed at the full strength of the *nmt1* promoter (Fig S2).

2. Remove the 3D effects of the cartoons of figure 1a, they just reduce visibility.

Our response: Fig 1A, Cartoons colorized for clarity.

3. Chang et al. PMID:7923372 already described the mating phenotype of *ras1* mutants, they should be cited.

Our response: Reference added.

4. Many references in the reference list are incomplete.

Our response: References checked

5. Figure 6 is referred to only at the discussion, even though it contains experimental data. I suggest that the discussion and the results are fused

Our response: As suggested, Results and Discussion combined.

I hope you will find that we have addressed the referees' comments to your satisfaction, and look forward to hearing from you in due course.

January 3, 2023

RE: Life Science Alliance Manuscript #LSA-2022-01755-TR

Dr. Shao-Win Wang
National Health Research Institutes
Institute of Molecular and Genomic Medicine
Keyan Road
Miaoli County 35053
Taiwan

Dear Dr. Wang,

Thank you for submitting your revised manuscript entitled "Protein S-palmitoylation regulates different stages of meiosis in *Schizosaccharomyces pombe*". We would be happy to publish your paper in Life Science Alliance pending final revisions necessary to meet our formatting guidelines.

- please add your supplementary figure legends to the main manuscript text and upload your supplementary figure files as separate single page files
- please add ORCID ID for corresponding author-you should have received instructions on how to do so
- please add the Twitter handle of your host institute/organization as well as your own or/and one of the authors in our system
- please consult our manuscript preparation guidelines <https://www.life-science-alliance.org/manuscript-prep> and make sure your manuscript sections are in the correct order and note that the Results and Discussion section should be two different sections

Figure Check:

- please add scale bars to Figure 3B, C, D, F and G; Figure 4A-D; G, I; Figure 5 B-D
- there is vertical line in the middle of the blot in Figure 4E. If this is the result of a splice between blots, please make the line more pronounced and mention this in the Figure 4 figure legend.

A. FINAL FILES:

B. MANUSCRIPT ORGANIZATION AND FORMATTING:

Sincerely,

Reviewer #1 (Comments to the Authors (Required)):

Dear Editor,

my previous comments were adequately addressed and the revised manuscript can be accepted for publication.

Best regards

Reviewer #3 (Comments to the Authors (Required)):

The authors have addressed all my concerns.

-please add your supplementary figure legends to the main manuscript text and upload your supplementary figure files as separate single page files

Our reponse: Supplementary figure legends are added to the main manuscript text, and each supplementary figure is uploaded as separate single file.

-please add ORCID ID for corresponding author-you should have received instructions on how to do so

Our reponse: ORCID ID provided.

-please add the Twitter handle of your host institute/organization as well as your own or/and one of the authors in our system

Our reponse: Our host institute and myself don't have Twitter account. @Pombase (curator of the Pombe community) is provided.

-Please consult our manuscript preparation guidelines <https://www.life-science-alliance.org/manuscript-prep> and make sure your manuscript sections are in the correct order and note that the Results and Discussion section should be two different sections

Our reponse: Results and Discussion are in two different sections

Figure Check:

-please add scale bars to Figure 3B, C, D, F and G; Figure 4A-D; G, I; Figure 5 B-D

Our reponse: Scale bars are added to the above Figures mentioned.

-there is vertical line in the middle of the blot in Figure 4E. If this is the result of a splice between blots, please make the line more pronounced and mention this in the Figure 4 figure legend.

Our reponse: The vertical line is due to the problem of our over 15 year very **old** CCD camera of UVP Biospectrum 500 Imaging System that we have not noticed before. It is not the result of a splice between blots. The original file is provided below. Please not that the line go all the way to the top beyond the blots. Thank you for bringing our attention to it. We will apply funding to replace it in the near future.

January 6, 2023

RE: Life Science Alliance Manuscript #LSA-2022-01755-TRR

Dr. Shao-Win Wang
National Health Research Institutes
Institute of Molecular and Genomic Medicine
No 35 Keyan Road
Miaoli County 35053
Taiwan

Dear Dr. Wang,

Thank you for submitting your Research Article entitled "Protein S-palmitoylation regulates different stages of meiosis in *Schizosaccharomyces pombe*". It is a pleasure to let you know that your manuscript is now accepted for publication in Life Science Alliance. Congratulations on this interesting work.

DISTRIBUTION OF MATERIALS:

Again, congratulations on a very nice paper. I hope you found the review process to be constructive and are pleased with how the manuscript was handled editorially. We look forward to future exciting submissions from your lab.

Sincerely,
